# `HypUC`: Hyperfine Uncertainty Calibration with Gradient-boosted Corrections for Reliable Regression on Imbalanced Electrocardiograms

**Uddeshya Upadhyay**[1]    **Sairam Bade**[1]    **Arjun Puranik**[1]    **Shahir Asfahan**[1]    **Melwin Babu**[1]

**Francisco Lopez-Jimenez**[3]    **Samuel J. Asirvatham**[3]

**Ashim Prasad**[1]    **Ajit Rajasekharan**[1]    **Samir Awasthi**[1,2]    **Rakesh Barve**[1,2]

[1]*Nference Inc.*    [2]*Anumana Inc.*    [3]*Mayo Clinic, USA*

**Reviewed on OpenReview:** *https://openreview.net/forum?id=0Xo9giEZWf*

## Abstract

The automated analysis of medical time series, such as the electrocardiogram (ECG), electroencephalogram (EEG), pulse oximetry, etc, has the potential to serve as a valuable tool for diagnostic decisions, allowing for remote monitoring of patients and more efficient use of expensive and time-consuming medical procedures. Deep neural networks (DNNs) have been demonstrated to process such signals effectively. However, previous research has primarily focused on classifying medical time series rather than attempting to regress the continuous-valued physiological parameters central to diagnosis. One significant challenge in this regard is the imbalanced nature of the dataset, as a low prevalence of abnormal conditions can lead to heavily skewed data that results in inaccurate predictions and a lack of certainty in such predictions when deployed. To address these challenges, we propose `HypUC`, a framework for imbalanced probabilistic regression in medical time series, making several contributions. (i) We introduce a simple kernel density-based technique to tackle the imbalanced regression problem with medical time series. (ii) Moreover, we employ a probabilistic regression framework that allows uncertainty estimation for the predicted continuous values. (iii) We also present a new approach to calibrate the predicted uncertainty further. (iv) Finally, we demonstrate a technique to use calibrated uncertainty estimates to improve the predicted continuous value and show the efficacy of the calibrated uncertainty estimates to flag unreliable predictions. `HypUC` is evaluated on a large, diverse, real-world dataset of ECGs collected from millions of patients, outperforming several conventional baselines on various diagnostic tasks, suggesting potential use-case for the reliable clinical deployment of deep learning models and a prospective clinical trial. Consequently, a hyperkalemia diagnosis algorithm based on `HypUC` is going to be the subject of a real-world clinical prospective study.

## 1 Introduction

Electrocardiogram (ECG) signals are widely used in the diagnosis and management of cardiovascular diseases. However, widely popular manual analysis of ECG signals can be time-consuming and subject to human error. Automated analysis of ECG signals using machine learning (ML) techniques has the potential to improve the accuracy and efficiency of ECG signal analysis. Moreover, it will open up the avenue for remote patient monitoring for managing the cardiovascular health of the patients at scale. Recent advances in ML have led to the development of various algorithms for the automated analysis of ECG signals. These include methods for ECG signal classification and diagnosis of specific car-

diovascular diseases (Hannun et al., 2019). Convolutional neural networks (CNNs) and recurrent neural networks (RNNs) are commonly used for these tasks (Siontis et al., 2021; Şentürk et al., 2018).

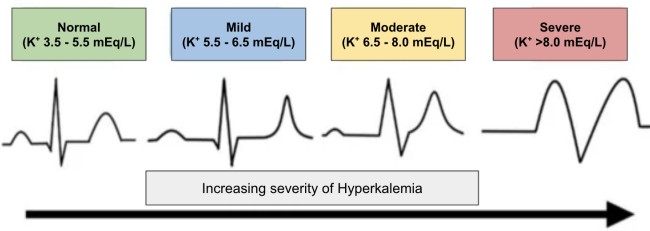

**Figure 1:** Severity of the Hyperkalemia often decides diagnosis. Hence, the binary classification is not sufficient.

Existing literature often focuses on classifying the medical time series (in one of the disease-related classes) (Siontis et al., 2021; Hannun et al., 2019) instead of regressing the continuous-valued physiological parameter on which the diagnosis is based (Von Bachmann et al., 2022). But often, solving the regression problem is more valuable than classification; for instance, consider the problem of measuring electrolytes like potassium in the blood. Excess of potassium (known as *Hyperkalemia*) can be fatal. Detecting if the patient has Hyperkalemia non-invasively by analyzing the ECG is desirable and has been studied previously (Galloway et al., 2019). However, a binary classifier does not distinguish between the different severity of Hyperkalemia as shown in Figure 1 that may decide the final diagnosis for the condition, i.e., severe/mild hyperkalemia requires different treatments.

One of the significant challenges in the real world for regressing the continuous-valued physiological parameter from medical time series is the imbalanced nature of the dataset, i.e., such datasets are often highly skewed if the prevalence of the abnormal condition is low. A heavily skewed dataset leads to a trained network that is inaccurate in the prediction. Moreover, a deterministic deep-learning model cannot quantify uncertainty in predictions at deployment. To address these limitations, we propose a new framework called `HypUC`, shown in Figure 2. It leverages ideas from probabilistic regression and density estimation to tackle uncertainty-aware imbalanced regression with medical time series. In particular, our method makes several contributions:

- We introduce a simple kernel density-based technique to tackle the imbalanced regression problem with medical time series, discussed in detail in Section 3.2.2, 3.3.

- Our method builds on probabilistic deep regression that allows uncertainty estimation for the predicted continuous values in the novel context of medical time series, presented in Section 3.2.3, 3.3.

- We propose a new approach for calibrating the predicted uncertainty in Section 3.3.2.

- Finally, we demonstrate the efficacy of the calibrated uncertainty estimates to improve the predicted continuous value and also flag unreliable predictions, as discussed in Section 3.3.3.

We study the efficacy of our methods on a large real-world dataset of ECGs (collected using millions of patients with diverse medical conditions) to predict various physiological parameters such as *Left Ventricular Ejection Fraction*, *Serum Potassium Level*, *Age*, *Survival*, etc.

## 2 Related Work

**Automated analysis of medical time series.** Deep learning techniques have been extensively used to analyze medical time series signals. For instance, the work by Schaekermann et al. (2020); Calisto et al. (2017; 2022; 2023); Enevoldsen et al. (2023) compare the performance of machine learning models on medical time series classification in the presence of ambiguity and improve the workflow of medical practitioners. There are also works that synthesize the personalized medical time series data, like glucose monitoring signal using generative models (Zhu et al., 2023). Similar to other time-series in helathcare, in the constext of electrocardiograms, one of the main areas of focus has been the use of convolutional neural networks (CNNs) for ECG signal *classification*. For instance, the studies by Rajpurkar et al. (2017); Hannun et al. (2019); Murugesan et al. (2018) used CNNs to classify ECG signals into normal and abnormal beats accurately. Another set of studies by Ebrahimi et al. (2020); Li et al. (2016) proposed a deep learning-based framework for arrhythmia

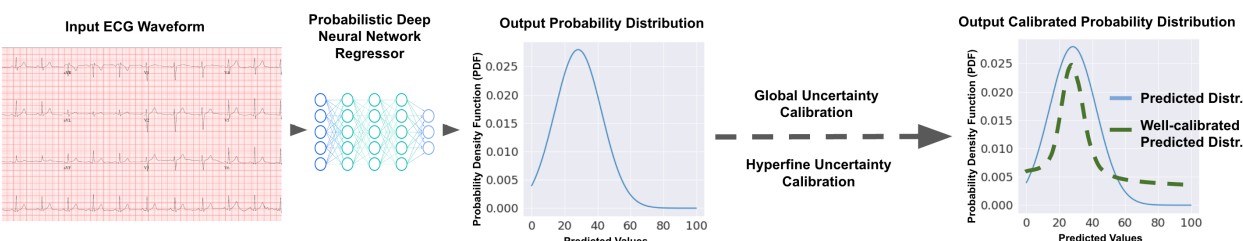

**Figure 2:** `HypUC`: The proposed framework processes the ECG waveform input to predict the parameters of the distribution (i.e., entire parametric distribution) representing the continuous value. The predicted distribution is subjected to global and hyperfine uncertainty calibration techniques to get a well-calibrated predictive distribution and uncertainty estimate as explained in Section 3.

detection using ECG signals. Moreover, some studies use ECGs to predict QRS complex location (Han et al., 2022; Camps et al., 2018), heart-rate (Xu et al., 2019; Şentürk et al., 2018), and cardiovascular diseases like myocardial infarction (Baloglu et al., 2019; Mahendran et al., 2021), hypertension (Martinez-Ríos et al., 2021) , atrial fibrillation (Siontis et al., 2021; Feeny et al., 2020), arrhythmia (Feeny et al., 2020; Pant et al., 2022), cardiac death (Ko et al., 2020), heart failure (Bos et al., 2021), etc.

However, it is crucial to note that many cardiovascular health-related diagnoses are based on *continuous value predictions*, for instance, hyper/hypo-kalemia is based on serum potassium concentration (Webster et al., 2002), low ventricular-ejection fraction is based on ejection fraction (Yao et al., 2020), etc. This underscores the need for newer methods that effectively tackle the *regression* problem. Moreover, in critical applications, like healthcare, it is desirable to quantify the uncertainty in the predictions made as it can prevent fatal decision-making and improve the overall outcomes (Chen et al., 2021; Tanno et al., 2021; Rangnekar et al., 2023; Upadhyay et al., 2021b). But, the above intersection has not been studied in the context of medical time series like ECGs. We explore this gap in the literature and provide effective solutions for the same.

## 3 Methodology

We first formulate the problem in Section 3.1. Preliminaries necessary to motivate the design choices for our framework (`HypUC`) are presented in Section 3.2.1 (on Imbalanced Regression), Section 3.2.2 (on Kernel Density Estimation), and Section 3.2.3 (on Uncertainty Estimation). In Section 3.3, we construct `HypUC`, that performs uncertainty-aware regression followed by a new calibration technique that significantly improves the quality of uncertainty estimates and utilizes a simple technique based on gradient-boosted decision trees to enhance decision-making using the predicted point estimate and well-calibrated uncertainty estimate.

### 3.1 Problem formulation

Let $\mathcal{D} = \{(\mathbf{x}_i, \mathbf{y}_i)\}_{i=1}^{N}$ be the training set with pairs from domain $\mathbf{X}$ and $\mathbf{Y}$ (i.e., $\mathbf{x}_i \in \mathbf{X}, \mathbf{y}_i \in \mathbf{Y}, \forall i$), where $\mathbf{X}, \mathbf{Y}$ lies in $\mathbb{R}^m$ and $\mathbb{R}$, respectively. While our proposed solution is valid for data of arbitrary input dimension, we present the formulation for medical time series (like ECGs) with applications that predict a desired physiological parameter helpful in making a diagnosis using ECG. Therefore, $(\mathbf{x}_i, \mathbf{y}_i)$ represents a pair of ECG and physiological quantity/parameter. For instance, to aid in diagnosing Hyperkalemia, $\mathbf{x}_i$ is ECG signal, and $\mathbf{y}_i$ is the serum potassium level (indicating the amount of potassium in the blood, which may be obtained from the blood test that serves as groundtruth labels for training).

We aim to learn a mapping, say $\boldsymbol{\Psi}(\cdot) : \mathbf{X} \to P(\mathbf{Y})$, that learns to map a given input ECG ($\mathbf{x}$) to a probability density function $P(\mathbf{y})$, indicating the probable values. We use a deep neural network (DNN) to parameterize $\boldsymbol{\Psi}$ with $\theta$ and solve an optimization problem to obtain optimal parameters $\theta^*$, as described in Section 3.3.

## 3.2 Preliminaries

This section presents the preliminary concepts highlighting the current challenges, existing literature, and some techniques used to construct our proposed framework, HypUC. In particular, Section 3.2.1 discusses the problem of imbalanced regression that is very typical in real-world clinical scenarios and some solutions to tackle the same using machine learning. Section 3.2.3 presents the existing framework often used to estimate the uncertainty in the predictions made by the deep neural network and the shortcomings associated with such methods. The culmination of ideas presented in this section helps us construct our method.

### 3.2.1 Imbalanced Regresison

Imbalanced regression refers to the problem of training a machine learning model for regression on a dataset where the target variable is imbalanced, meaning that specific values appear more often than others. This is typical for clinical datasets where different medical conditions appear with different prevalences.

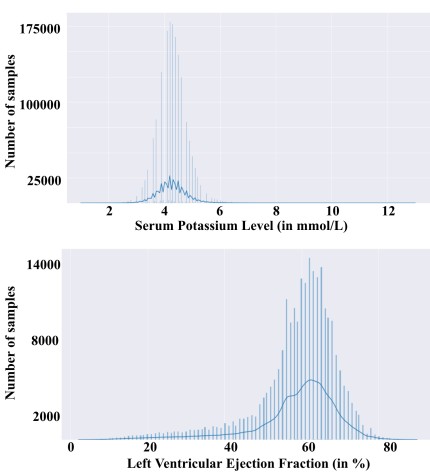

**Figure 3:** Imbalanced distribution of target values (top) Serum Potassium Levels and (bottom) Ejection Fraction in real-world medical datasets.

For instance, Figure 3 shows the density and the distribution of Serum Potassium Levels (i.e., the concentration of potassium in the blood) and left ventricular ejection fraction (i.e., the volume of blood pumped out of the left ventricle of the heart relative to its total volume) for a real-world dataset. We notice that most of the patients have serum potassium level in the range of 3.5 to 5.5 (which is often accepted as the range for normal potassium level). However, it is crucial for machine learning models to work well in the high potassium levels (defined above 5.5) that occur rarely, i.e., the population density in the region is very low. A similar trend is observed for LVEF, where most of the data is concentrated in the healthy range (between 40-60%), but low-LVEF regions (defined below 40%), which is essential for the model to identify, are sparsely present in the dataset. Imbalanced datasets can present challenges for training regression models, as the model may be biased towards the densely populated samples. Works by Yang et al. (2021); Steininger et al. (2021) proposes distribution smoothing for both labels and features, which explicitly acknowledges the effects of nearby targets and calibrates both labels and learned feature distributions lead to better point estimates in sparsely populated regions. However, the method does not explore the application to healthcare, where uncertainty quantification is critical, and the extension to incorporate uncertainty estimation in a similar framework is non-trivial. Our proposed HypUC proposes a method to tackle imbalanced regression using deep neural networks that also incorporates a novel technique for calibrated uncertainty quantification.

### 3.2.2 Kernel Density Estimation

Kernel density estimation (KDE) (Terrell & Scott, 1992; Chen, 2017) is the application of kernel smoothing for probability density estimation. Given a set of i.i.d. samples $\{\mathbf{y}_1, \mathbf{y}_2...\mathbf{y}_n\}$ drawn from some unknown density function $\mathbf{f}$. The kernel density estimate provides an estimate of the shape of $\mathbf{f}$. This can be leveraged to train deep regression models with imbalanced dataset by estimating the density for each target value in the training set and use this density to create a weighting scheme to be applied with the loss for that sample. This approach can help to give more

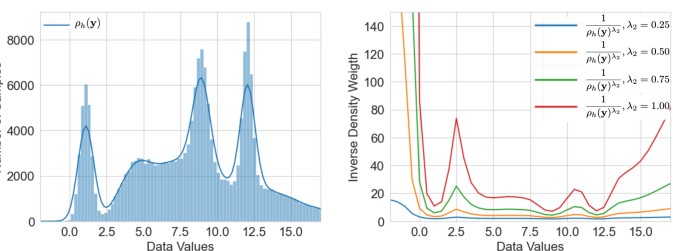

**Figure 4:** KDE and related weights for imbalanced targets. (left) Example of imbalanced target datasets with KDE. (right) Inverse of KDE with exponent used as weights for the loss term. Higher exponent leads to sharper weights.

importance to samples with rare target values, which are often more difficult to predict. This is achieved by estimating the density as:

$$\rho_h(\mathbf{y}) = \frac{1}{n}\sum_{i=1}^{n}\mathcal{K}_h(\mathbf{y}-\mathbf{y}_i) = \frac{1}{nh}\sum_{i=1}^{n}\mathcal{K}\left(\frac{\mathbf{y}-\mathbf{y}_i}{h}\right) \tag{1}$$

where $\mathcal{K}$ is the kernel — a non-negative function — and $h > 0$ is a smoothing parameter called the bandwidth. Scaled kernel $\mathcal{K}_h$ is defined as, $\mathcal{K}_h(\mathbf{y}) = \frac{1}{h}\mathcal{K}(\frac{\mathbf{y}}{h})$, and $\mathcal{K}$ is often set to standard normal density function. The weighting scheme can be designed by leveraging the fact that lower-density samples must receive higher weights and vice-versa, a simple technique to achieve the same is to design the weight for target value $\mathbf{y}$ to be $\mathbf{w}(\mathbf{y}) = \frac{1}{\rho_h(\mathbf{y})^{\lambda_2}}$. Figure 4-(left) shows an example of imbalanced target distribution and Figure 4-(right) shows the designed weighting scheme, we note that hyperparameter $\lambda_2$ controls the sharpness of the weights.

### 3.2.3 Uncertainty Estimation

The works in (Kendall & Gal, 2017; Laves et al., 2020b; Upadhyay et al., 2022; 2023; Sudarshan et al., 2021) discuss the key techniques to estimate the *irreducible* (i.e., aleatoric) uncertainty, and *reducible* (i.e., epistemic) uncertainty, using deep neural networks and also highlight the importance of capturing aleatoric uncertainty in the presence of large datasets. For an arbitrary deep neural network (DNN), say $\mathbf{\Phi}(\cdot;\zeta)$, parameterized by learnable parameters $\zeta$, that learns a probabilistic mapping from input domain $\mathbf{X}$ and output domain $\mathbf{Y}$ (as defined in Section 3), i.e., $\mathbf{\Phi}(\cdot;\zeta):\mathbf{X}\to\mathbf{Y}$. To estimate the aleatoric uncertainty, the model must estimate the parameters of the conditional probability distribution, $\mathcal{P}_{Y|X}$, which is then used to maximize the likelihood function. This distribution is assumed to be parametrized by set of parameters $\{\hat{\mathbf{y}}_i,\hat{\nu}_{i1},\hat{\nu}_{i2},\dots\hat{\nu}_{ik},\}$ which we need to estimate (e.g., for Laplace distribution the set of parameters is given by location and scale, $\{\mu,b\}$ (Ilg et al., 2018), for Gaussian distribution the set of parameters is given by mean and standard-deviation, i.e, $\{\mu,\sigma\}$ (Kendall & Gal, 2017), for generalized Gaussian the set of parameters is given by location, scale, and shape parameters $\{\mu,\alpha,\beta\}$ (Upadhyay et al., 2021a)). That is, for an input $\mathbf{x}_i$, the model produces a set of parameters representing the output given by, $\{\hat{\mathbf{y}}_i,\hat{\nu}_{i1},\hat{\nu}_{i2},\dots\hat{\nu}_{ik}\} := \mathbf{\Phi}(\mathbf{x}_i;\zeta)$, that characterizes the distribution $\mathcal{P}_{Y|X}(\mathbf{y};\{\hat{\mathbf{y}}_i,\hat{\nu}_{i1},\hat{\nu}_{i2},\dots\hat{\nu}_{ik}\})$, such that $\mathbf{y}_i \sim \mathcal{P}_{Y|X}(\mathbf{y};\{\hat{\mathbf{y}}_i,\hat{\nu}_{i1},\hat{\nu}_{i2},\dots\hat{\nu}_{ik}\})$. The likelihood $\mathscr{L}(\zeta;\mathcal{D}) := \prod_{i=1}^{N}\mathcal{P}_{Y|X}(\mathbf{y}_i;\{\hat{\mathbf{y}}_i,\hat{\nu}_{i1},\hat{\nu}_{i2},\dots\hat{\nu}_{ik}\})$ is then maximized to estimate the optimal parameters of the network. Moreover, the distribution $\mathcal{P}_{Y|X}$ is often chosen such that uncertainty can be computed using a closed form solution $\mathscr{F}$ of estimated parameters, i.e.,

$$\{\hat{\mathbf{y}}_i,\hat{\nu}_{i1},\hat{\nu}_{i2},\dots\hat{\nu}_{ik}\} := \mathbf{\Phi}(\mathbf{x}_i;\zeta) \text{ with } \zeta^* := \underset{\zeta}{\operatorname{argmax}}\,\mathscr{L}(\zeta;\mathcal{D}) = \underset{\zeta}{\operatorname{argmax}}\prod_{i=1}^{N}\mathcal{P}_{Y|X}(\mathbf{y}_i;\{\hat{\mathbf{y}}_i,\hat{\nu}_{i1},\hat{\nu}_{i2},\dots\hat{\nu}_{ik}\})$$

$$\tag{2}$$

$$\text{Uncertainty}(\hat{\mathbf{y}}_i) = \mathscr{F}(\hat{\nu}_{i1},\hat{\nu}_{i2},\dots\hat{\nu}_{ik}) \tag{3}$$

It is common to use a *heteroscedastic* Gaussian distribution for $\mathcal{P}_{Y|X}$ (Kendall & Gal, 2017; Laves et al., 2020b), in which case $\mathbf{\Phi}(\cdot;\zeta)$ is designed to predict the *mean* and *variance* of the Gaussian distribution, i.e., $\{\hat{\mathbf{y}}_i,\hat{\sigma}_i^2\} := \mathbf{\Phi}(\mathbf{x}_i;\zeta)$, and the predicted *variance* itself can be treated as uncertainty in the prediction. The optimization problem becomes,

$$\zeta^* = \underset{\zeta}{\operatorname{argmax}}\prod_{i=1}^{N}\frac{1}{\sqrt{2\pi\hat{\sigma}_i^2}}e^{-\frac{|\hat{\mathbf{y}}_i-\mathbf{y}_i|^2}{2\hat{\sigma}_i^2}} = \underset{\zeta}{\operatorname{argmin}}\sum_{i=1}^{N}\frac{|\hat{\mathbf{y}}_i-\mathbf{y}_i|^2}{2\hat{\sigma}_i^2}+\frac{\log(\hat{\sigma}_i^2)}{2} \tag{4}$$

$$\text{Uncertainty}(\hat{\mathbf{y}}_i) = \hat{\sigma}_i^2. \tag{5}$$

Work by Laves et al. (2020a) discusses that uncertainty estimated using above technique is not well calibrated and provides a remedy for the same. They demonstrate that learning a scaling factor for the estimated uncertainty in a post-hoc optimization step on validation data (i.e., after training) help improve the calibration. However, this technique learns a single scaling factor, that is not optimal for imbalanced datasets (e.g., clinical datasets as discussed in Section 3.2.1). We propose to alleviate this problem by proposing a hyperfine uncertainty calibration scheme as discussed in Section 3.3.2.

### 3.3 Building `HypUC`

In this section, we leverage the preliminary concepts as discussed above to construct our proposed `HypUC`. We explain the building blocks for our framework, which consists of three major steps: (i) Optimization for improved continuous-value point and uncertainty estimation, (ii) Post-training hyperfine uncertainty calibration, (iii) Improved decision-making with an ensemble of gradient-boosted learners.

#### 3.3.1 Optimization for Improved Continuous-value Point and Uncertainty Estimation

As described in Section 3.1, we aim to learn the estimation function $\mathbf{\Psi}(\cdot; \theta)$. Here we design the network $\mathbf{\Psi}(\cdot; \theta)$ to model the output as a Gaussian distribution as thoroughly discussed in previous works (Kendall & Gal, 2017; Laves et al., 2020b; Upadhyay et al., 2021b). That is, the DNN $\mathbf{\Psi}$ takes the medical time series as the input and predicts the probability density function parameters for the output, i.e., $\mathbf{\Psi}(\mathbf{x}; \theta) = \{\hat{\mathbf{y}}, \hat{\sigma}\} = \{[\mathbf{\Psi}(\mathbf{x}_i; \theta)]_{\hat{y}}, [\mathbf{\Psi}(\mathbf{x}_i; \theta)]_{\hat{\sigma}}\}$. Where $[\mathbf{\Psi}(\mathbf{x}_i; \theta)]_{\hat{y}}$ is the $\hat{\mathbf{y}}_i$ estimate given by the network and $[\mathbf{\Psi}(\mathbf{x}_i; \theta)]_{\hat{\sigma}}$ is the $\hat{\sigma}_i$ estimate given by the network. We propose to learn the optimal parameters of the DNN, $\theta^*$, by:

$$\theta^* = \underset{\theta}{\operatorname{argmin}} \frac{1}{N} \sum_{i=1}^{N} \left\{ \lambda_1 * \left( \frac{|([\mathbf{\Psi}(\mathbf{x}_i; \theta)]_{\hat{y}} - \mathbf{y}_i)|}{\rho_h(\mathbf{y}_i)^{\lambda_2}} \right) + \lambda_3 * \left( \frac{|([\mathbf{\Psi}(\mathbf{x}_i; \theta)]_{\hat{y}} - \mathbf{y}_i)|^2}{[\mathbf{\Psi}(\mathbf{x}_i; \theta)]_{\hat{\sigma}}^2} + \log \left( [\mathbf{\Psi}(\mathbf{x}_i; \theta)]_{\hat{\sigma}}^2 \right) \right) \right\}. \quad (6)$$

Where $\rho_h(\mathbf{y})$ is the Gaussian kernel density estimate (fit on the training labels) for the target value $\mathbf{y}$. $\lambda_1, \lambda_2, \lambda_3, h$ are the hyperparameters, controlling the strength of imbalance correction and its relative contribution with respect to uncertainty-based regression terms as detailed below.

In the above optimization, the term $\left( \frac{|([\mathbf{\Psi}(\mathbf{x}_i; \theta)]_{\hat{y}} - \mathbf{y}_i)|}{\rho_h(\mathbf{y}_i)^{\lambda_2}} \right)$ computes the discrepancy between the continuous value prediction and the groundtruth, where the discrepancy is weighed down if the groundtruth label is present in abundance, i.e., density $\rho_h(\mathbf{y})$ is high. The strength of the weighting term is controlled by the hyperparameters $h, \lambda_2$. The discrepancy is weighed up if the label density is low, addressing the imbalanced nature of the dataset. The term, $\left( \frac{|([\mathbf{\Psi}(\mathbf{x}_i; \theta)]_{\hat{y}} - \mathbf{y}_i)|^2}{[\mathbf{\Psi}(\mathbf{x}_i; \theta)]_{\hat{\sigma}}^2} + \log \left( [\mathbf{\Psi}(\mathbf{x}_i; \theta)]_{\hat{\sigma}}^2 \right) \right)$, in the above optimization, is the negative-log-likelihood of heteroscedastic Gaussian distribution assumed for each prediction. The hyperparameters $\lambda_1, \lambda_3$ control the relative contribution of the density-based discrepancy term and the heteroscedastic negative-log-likelihood term that learns to estimate the uncertainty.

#### 3.3.2 Post-training Hyperfine Uncertainty Calibration

Once the DNN is trained, i.e., the optimal parameters $\theta^*$ have been found as described in the above optimization step. The trained DNN, $\mathbf{\Psi}(; \theta^*)$, is capable of estimating both the target continuous value and the uncertainty in the prediction. However, the uncertainty estimates obtained are biased and are not well calibrated (Laves et al., 2020a; Bishop & Nasrabadi, 2006; Hastie et al., 2009). Therefore, we propose a new technique to calibrate the uncertainty estimates. We first find a global optimal scaling factor $s^*$ by solving the following optimization problem on the validation set $\mathcal{D}_{valid} = \{(\mathbf{x}_i, \mathbf{y}_i)\}_{i=1}^{M}$, where the parameters of the DNN are fixed to the optimal obtained above (i.e., $\theta^*$) as described in (Laves et al., 2020a),

$$s^* = \underset{s}{\operatorname{argmin}} \left( M \log(s) + \frac{1}{2s^2} \sum_{i=1}^{M} \frac{|([\mathbf{\Psi}(\mathbf{x}_i; \theta^*)]_{\hat{y}} - \mathbf{y}_i)|^2}{[\mathbf{\Psi}(\mathbf{x}_i; \theta^*)]_{\hat{\sigma}}^2} \right) \quad (7)$$

We then run the model in inference mode and generate the estimates for all the samples in the validation set, i.e., we get a new augmented validation set, $\mathcal{D}_{valid-aug} = \{\mathbf{x}_i, \mathbf{y}_i, \hat{\mathbf{y}}_i, \hat{\sigma}_i^2\}_{i=1}^{M}$. We then create hyperfine bins of uniform length spanning the entire range of predicted continuous-values, i.e., we create $b = int \left( \frac{\mathbf{y}_{max} - \mathbf{y}_{min}}{\delta} \right)$ bins, where $\delta << \mathbf{y}_{max} - \mathbf{y}_{min}$ leading to hyperfine bins, where $\mathbf{y}_{max} = max(\{\mathbf{y}_i\}_{i=1:M})$ and $\mathbf{y}_{min} = min(\{\mathbf{y}_i\}_{i=1:M})$. The $n^{th}$ bin being $B_n$ spanning $[\mathbf{y}_{min} + (n-1)\delta, \mathbf{y}_{min} + n\delta]$. Each sample in the validation set then belongs to one of the bins. We then find a bin-wise scaling factor for each of the above hyperfine bins by solving an optimization problem. The bin-wise scaling factor for bin $B_n$, $\eta_{B_n}$, is given by,

$$\eta_{B_n} = \underset{\eta}{\operatorname{argmin}} \left( \frac{len(\{|\hat{\mathbf{y}}_i - \mathbf{y}_i| < \eta s^* \hat{\sigma}_i\}_{\hat{\mathbf{y}}_i \in B_n})}{len(B_n)} > \xi \right) \quad (8)$$

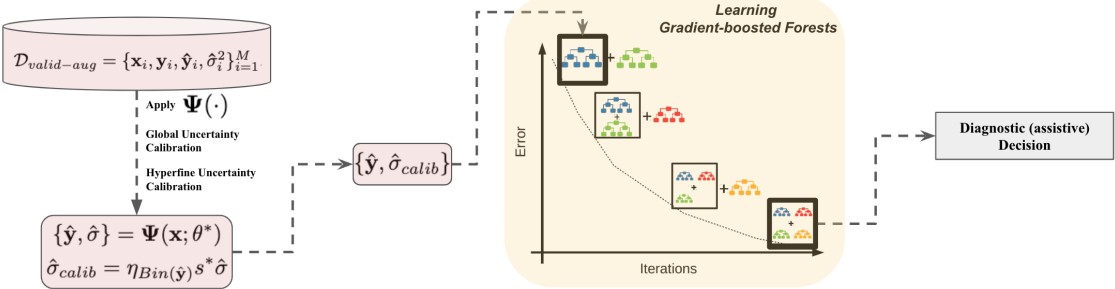

**Figure 5:** Gradient-boosted based ensemble of decision trees incorporating calibrated uncertainty (via both global and hyperfine calibration) and point estimate to help in diagnostic decision making.

Where, $len(\cdot)$ is the function that gives the number of elements in the input set and hyperparameter $\xi$ represents the minimum acceptable fraction in each bin where the discrepancy between the prediction and the groundtruth (i.e., $|\hat{\mathbf{y}} - \mathbf{y}|$) is captured by the scaled uncertainty estimates (i.e., $\eta s^* \hat{\sigma}_i$). The above scaling factors allow us to compute the well-calibrated uncertainty estimates for each prediction. Given input $\mathbf{x}$, the continuous value prediction ($\hat{\mathbf{y}}$) and the calibrated uncertainty ($\hat{\sigma}_{calib}$) are given by,

$$\{\hat{\mathbf{y}}, \hat{\sigma}\} = \mathbf{\Psi}(\mathbf{x}; \theta^*) \text{ and } \hat{\sigma}_{calib} = \eta_{Bin(\hat{\mathbf{y}})} s^* \hat{\sigma} \tag{9}$$

Where, $Bin(\hat{\mathbf{y}})$ is the hyperfine bin to which $\hat{\mathbf{y}}$ belongs $\eta_{Bin(\hat{\mathbf{y}})}$ is the bin-wise scaling factor of the bin.

### 3.3.3 Improved Decision-making with an Ensemble of Gradient-boosted Learners

Medical professionals often diagnose the condition based on continuous value lying in a certain standardized range (Galloway et al., 2019; Bachtiger et al., 2022). We propose a technique that utilizes both the predicted continuous value and calibrated uncertainty estimate, as obtained in the above step, to infer the standardized range (that is better than relying on just the continuous-value point estimate). The proposed technique uses predictions (both $\hat{\mathbf{y}}$ and $\hat{\sigma}_{calib}$) from the trained DNN ($\mathbf{\Psi}$) as input features to an ensemble of decision trees that uses a gradient-boosting technique to learn a random forest (represented by $\mathbf{\Upsilon}$ with $N_{trees}$ trees and $d_{tree}$ as the maximum possible depth for each tree) on dataset $\mathcal{D}_{train} \cup \mathcal{D}_{valid}$, that infers the standardized-range/class in which the predicted continuous-value belongs, accounting for the uncertainty in the predictions,

$$\text{Class}(\mathbf{x}) = \mathbf{\Upsilon}\left([\mathbf{\Psi}(\mathbf{x}; \theta^*)]_{\hat{y}}, \eta_{Bin([\mathbf{\Psi}(\mathbf{x};\theta^*)]_{\hat{y}})} s^* [\mathbf{\Psi}(\mathbf{x}; \theta^*)]_{\hat{\sigma}}\right). \tag{10}$$

In the above, note that the regression model also allows classification. This can be achieved by imposing interpretation rules on the predicted continuous values (either explicitly by thresholding the continuous value predictions for different classes or implicitly by learning a classifier on top of predicted continuous values and other features, as described in this section). Therefore, this approach could potentially allow medical practitioners to not only know about the severity of the condition (indicated by the continuous value), but also use it to infer the broad category (by deriving the class disease class using continuous value as described).

## 4 Experiments and Results

### 4.1 Tasks and Datasets

Our dataset is curated at `Mayo Clinic, USA` and consists of over *2.5 million* patients, with over *8.9 million* 12-lead ECGs coming from different types of hardware, covering a wide range of medical conditions, age, and demographic variables. The data collection is from the years 1980 to 2020. The retrospective ECG-AI research described herein has been approved and determined to be minimal risk by the Mayo Clinic Institutional Review Board (IRB 18-008992).

### 4.1.1 Estimating Age from ECG

Estimating age from an electrocardiogram (ECG) has gained increasing attention in recent years due to the potential for non-invasive age estimation to aid in diagnosing and treating age-related conditions. Moreover, ECGs have been shown to indicate cardiovascular age, which is helpful in determining the overall cardiovascular health (van der Wall et al., 2022; Chang et al., 2022; Baek et al., 2023). Previous works like (Attia et al., 2019) have shown that using deep learning, it is possible to analyze the ECG signals such that the neural network can identify patterns in the ECG waveform that change with age, which can be used to estimate an individual's cardiovascular age. For our experiments, we curated a diverse dataset of 8.8 million ECGs (along with the patient's age). We used 6.7/0.75/1.35 million for train/val/test datasets.

### 4.1.2 Estimating Survival (time-to-death) from ECG

Estimating survival, or time to death, from an electrocardiogram (ECG) is a relatively new field of study that has gained momentum in recent years as it may aid in the treatment and management of life-threatening conditions (Diller et al., 2019; Tripoliti et al., 2017). We curate the data to consist only of patients for which the death has been recorded. The time interval from ECG recording to death for such patients provides groundtruth for survival (time-to-death). In total, there were 3 million ECGs with the corresponding time-to-death, we used 2.3/0.26/0.44 million for train/val/test sets.

### 4.1.3 Estimating Level of Potassium in Blood from ECG: Detecting Hyperkalemia

Serum potassium level measures the amount of potassium in the blood. Potassium is an electrolyte essential for properly functioning the body's cells, tissues, and organs. It helps regulate the balance of fluids in the body and maintain normal blood pressure, among other functions. The normal range for serum potassium levels is typically between 3.5-5.0 millimoles per liter (mmol/L). Levels below 3.5 mmol/L are considered to be low (*hypokalemia*), and levels above 5.0 mmol/L are considered to be high (*hyperkalemia*). Both low and high potassium levels can have serious health consequences. While a blood test is used to measure the serum potassium level, recent works have evaluated machine learning techniques to infer the potassium level using ECGs. This can play a critical role in remote patient monitoring (Galloway et al., 2019; 2018; Feeny et al., 2020; Ahn et al., 2022). In total, there are over 4 million ECGs with the corresponding serum potassium level (derived using blood test). We used 2.1/0.4/1.7 million for train/val/test sets.

### 4.1.4 Estimating Left Ventricular Ejection Fraction from ECG: Detecting low LVEF

Low *left ventricular ejection fraction* (LVEF) is a condition in which the left ventricle of the heart is unable to pump a sufficient amount of blood out of the heart and into the body. The left ventricle is the main pumping chamber of the heart and is responsible for pumping oxygenated blood to the rest of the body. LVEF is typically measured as a percentage, with a normal range being between $55 - 70\%$. A LVEF below $40\%$ is generally considered to be low and may indicate heart failure. LVEF can be evaluated through various non-invasive tests, including an echocardiogram, a nuclear stress test. But ECG can still provide valuable information about heart function and can be used in conjunction with other tests to assess LVEF and diagnose heart conditions (Noseworthy et al., 2020; Vaid et al., 2022). In our experiments, we leverage DNNs to infer the LVEF - this can help reduce the costs associated with overall diagnosis and treatment by reducing the need for more expensive Echocardiogram. In total, there are over 0.6 million ECGs with the corresponding LVEF (derived using Echo). We used 0.3/0.1/0.2 million for train/val/test sets.

## 4.2 Compared Methods and Training details

To perform a comparative study with 12-lead ECG time series, we create a backbone DNN architecture adapting ResNets into a 1D version (with 1D convolutional layers) as discussed in previous works (Attia et al., 2019). The DNN backbone architecture is used to create a class of (i) deterministic regression models, (ii) class of probabilistic regression models, and (iii) class of classification models for some of the tasks. For the probabilistic regression, we append the backbone architecture with two smaller feed-forward heads to estimate the parameters of the predictive distribution as described in (Kendall & Gal, 2017; Upadhyay et al.,

2022). The deterministic regression model is trained with (i) L2 loss, termed as `Regres.-L2`, (ii) L1 loss termed as `Regres.-L1`, (iii) We incorporate the density-based weights to modulate the loss terms, we call this `Regres.-L1-KDEw`. We train two variants of probabilistic regression models (i) the standard *heteroscedastic Gaussian* model that estimates the mean and variance for the prediction and trained to maximize the likelihood as described in (Von Bachmann et al., 2022), called `Regres.-w.-U` and (ii) Our proposed model `HypUC`, that incorporates a novel calibration technique along with gradient-boosted corrections. For serum potassium level estimation (and LVEF estimation), we also compare with a binary classifier that predicts whether the given ECG has Hyperkalemia (and low LVEF for LVEF estimation). All the regression models are trained on standardized target values, i.e., $(target - \mu_{target})/(\sigma_{target})$. We also present the regression models trained with standardized log targets, i.e., $(\log(target) - \mu_{\log(target)})/(\sigma_{\log(target)})$. We train all the models using Adam optimizer (Kingma & Ba, 2014), with $(\beta_1, \beta_2)$ set to (0.9,0.999). The learning rate used to train models was $1e-4$. For `HypUC`, the hyper-parameters $\{\lambda_1, h, \lambda_2, \lambda_3\}$ are set to (i) $\{1, 0.7, 0.8, 1e-3\}$ for Age estimation. (ii) $\{1, 0.5, 0.5, 1e-4\}$ for Survival estimation. (iii) $\{1, 1, 0.2, 1e-4\}$ for Serum potassium estimation. (iv) $\{1, 1.5, 0.1, 1e-4\}$ for Left Ventricular Ejection Fraction (LVEF) estimation.

| Tasks | Methods | MSE ↓ | MAE ↓ | Spear. Corr. ↑ | Pears. Corr. ↑ | UCE ↓ | NLL ↓ | $\mathcal{I}_{0.95}$ ↑ | $<len(\mathcal{I}_{0.95})>$ ↓ | AUC ↑ | Sensiti. ↑ | Specifi. ↑ | PPV ↑ | NPV ↑ |
|---|---|---|---|---|---|---|---|---|---|---|---|---|---|---|
| Survival Estimation | Regres.-L2 | 100.12 | 8.49 | 0.30 | 0.30 | NA | NA | NA | NA | NA | NA | NA | NA | NA |
| | Regres.-L1 | 73.21 | 7.78 | 0.38 | 0.41 | NA | NA | NA | NA | NA | NA | NA | NA | NA |
| | Regres.-L1 (log) | 64.33 | 6.11 | 0.51 | 0.54 | NA | NA | NA | NA | NA | NA | NA | NA | NA |
| | Regres.-L1-KDEw | 67.97 | 6.30 | 0.51 | 0.53 | NA | NA | NA | NA | NA | NA | NA | NA | NA |
| | Regres.-w.-U | 84.62 | 8.13 | 0.35 | 0.38 | 2.37 | 5.89 | 0.89 | 20.5 | NA | NA | NA | NA | NA |
| | Regres.-w.-U (log) | 82.23 | 8.06 | 0.36 | 0.38 | 2.14 | 5.22 | 0.90 | 19.2 | NA | NA | NA | NA | NA |
| | HypUC | **54.62** | **5.38** | **0.56** | **0.61** | **0.58** | **3.45** | **0.93** | **13.4** | NA | NA | NA | NA | NA |
| | HypUC (log) | 55.77 | 5.52 | 0.52 | 0.59 | 0.69 | 3.62 | 0.91 | 15.6 | NA | NA | NA | NA | NA |
| Age Estimation | Regres.-L2 | 151.69 | 9.47 | 0.58 | 0.60 | NA | NA | NA | NA | NA | NA | NA | NA | NA |
| | Regres.-L1 | 116.52 | 7.93 | 0.73 | 0.75 | NA | NA | NA | NA | NA | NA | NA | NA | NA |
| | Regres.-L1 (log) | 101.58 | 8.13 | 0.66 | 0.68 | NA | NA | NA | NA | NA | NA | NA | NA | NA |
| | Regres.-L1-KDEw | 99.77 | 7.64 | 0.78 | 0.81 | NA | NA | NA | NA | NA | NA | NA | NA | NA |
| | Regres.-w.-U | 136.27 | 8.22 | 0.68 | 0.72 | 13.32 | 7.84 | 0.88 | 17.6 | NA | NA | NA | NA | NA |
| | Regres.-w.-U (log) | 138.23 | 8.56 | 0.68 | 0.71 | 12.95 | 7.23 | 0.91 | 17.1 | NA | NA | NA | NA | NA |
| | HypUC | **74.9** | **6.70** | **0.84** | **0.88** | **1.06** | **3.70** | **0.92** | **11.3** | NA | NA | NA | NA | NA |
| | HypUC (log) | 76.5 | 7.13 | 0.81 | 0.84 | 1.37 | 3.93 | 0.91 | 14.8 | NA | NA | NA | NA | NA |
| Serum Potassium Estimation | Binary-Classifier | NA | NA | 0.39 | 0.41 | NA | NA | NA | NA | 0.87 | 0.74 | 0.85 | 0.10 | 0.99 |
| | Regres.-L2 | 0.26 | 0.52 | 0.51 | 0.54 | NA | NA | NA | NA | 0.84 | 0.70 | 0.86 | 0.10 | 0.99 |
| | Regres.-L1 | 0.19 | 0.34 | 0.52 | 0.54 | NA | NA | NA | NA | 0.86 | 0.70 | **0.87** | 0.09 | 0.99 |
| | Regres.-L1 (log) | 0.21 | 0.39 | 0.48 | 0.50 | NA | NA | NA | NA | 0.85 | 0.71 | 0.84 | 0.10 | 0.99 |
| | Regres.-L1-KDEw | 0.19 | 0.33 | 0.52 | 0.55 | NA | NA | NA | NA | 0.86 | 0.72 | 0.85 | **0.11** | 0.99 |
| | Regres.-w.-U | 0.23 | 0.38 | 0.51 | 0.55 | 1.83 | 8.76 | 0.89 | 5.1 | 0.86 | 0.71 | 0.86 | 0.10 | 0.99 |
| | Regres.-w.-U (log) | 0.23 | 0.37 | 0.51 | 0.55 | 1.97 | 8.82 | 0.89 | 5.0 | 0.86 | 0.71 | 0.86 | 0.10 | 0.99 |
| | HypUC | 0.20 | **0.32** | 0.52 | 0.55 | **0.41** | **3.12** | **0.94** | **3.1** | **0.89** | **0.77** | 0.86 | 0.10 | **0.99** |
| | HypUC (log) | 0.21 | 0.36 | 0.52 | 0.57 | 0.49 | 3.78 | 0.93 | 3.8 | 0.88 | 0.75 | 0.84 | 0.10 | 0.99 |
| Ejection Fraction Estimation | Binary-Classifier | NA | NA | 0.28 | 0.32 | NA | NA | NA | NA | 0.91 | 0.77 | 0.87 | 0.09 | 0.98 |
| | Regres.-L2 | 171.37 | 13.56 | 0.37 | 0.39 | NA | NA | NA | NA | 0.90 | 0.75 | 0.84 | 0.09 | 0.98 |
| | Regres.-L1 | 153.66 | 11.78 | 0.41 | 0.42 | NA | NA | NA | NA | 0.91 | 0.76 | 0.86 | 0.10 | 0.98 |
| | Regres.-L1 (log) | 138.32 | 10.23 | 0.44 | 0.49 | NA | NA | NA | NA | 0.91 | 0.75 | 0.86 | 0.10 | 0.98 |
| | Regres.-L1-KDEw | 141.62 | 10.71 | 0.41 | 0.43 | NA | NA | NA | NA | 0.91 | 0.75 | 0.87 | 0.09 | 0.99 |
| | Regres.-w.-U | 165.81 | 12.24 | 0.41 | 0.44 | 10.56 | 9.83 | 0.89 | 35.1 | 0.90 | 0.74 | 0.85 | 0.10 | 0.99 |
| | Regres.-w.-U (log) | 150.33 | 11.25 | 0.40 | 0.43 | 8.11 | 8.36 | 0.90 | 30.3 | 0.90 | 0.75 | 0.85 | 0.10 | 0.99 |
| | HypUC | **133.26** | **10.21** | **0.43** | **0.45** | **1.68** | **4.28** | **0.91** | **26.2** | **0.93** | **0.77** | **0.87** | 0.10 | **0.99** |
| | HypUC (log) | 139.32 | 11.16 | 0.42 | 0.44 | 2.19 | 4.84 | 0.91 | 29.6 | 0.92 | 0.75 | 0.86 | 0.10 | 0.99 |

**Table 1:** Quantitative results on 4 different tasks (Survival, Age, Serum Potassium, and Left Ventricular Ejection Fraction Estimation) showing the performance of various deterministic and probabilistic regression and classification models in terms of Mean Squared Error (MSE), Mean Absolute Error (MAE), Spearman Correlation (Spear. Corr.), Pearson Correlation (Pear. Corr.), Uncertainty Calibration Error (UCE), Negative Log-likelihood (NLL), $\mathcal{I}_{0.95}$, $<len(\mathcal{I}_{0.95})>$, AUC, Sensitivity, Specificity, Positive/Negative Predictive Values PPV/NPV. If, for a task/method certain metric is not applicable, we show NA.

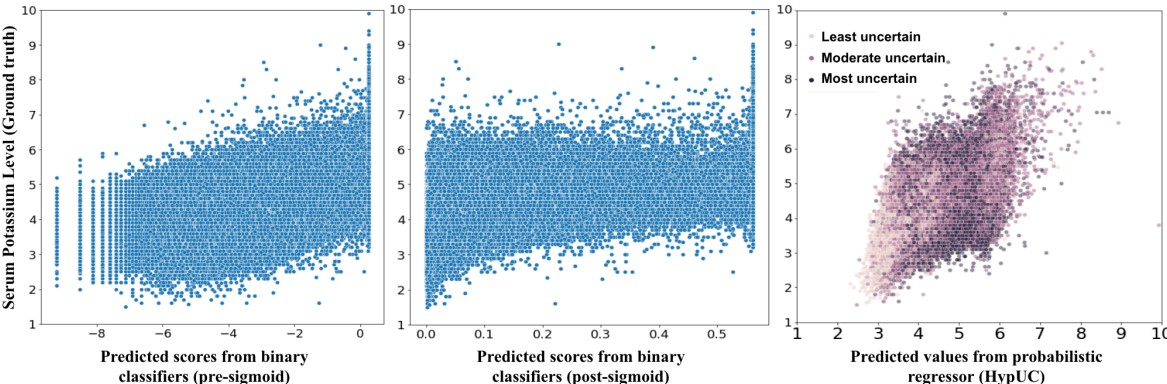

**Figure 6:** Serum Potassium Estimation model score correlation with groundtruth Serum Potassium levels. (left) Model scores (pre-activation) for the binary classifier. (middle) Model scores (post-activation) for the binary classifier. (right) Model scores for the proposed `HypUC`. The proposed `HypUC` not only produces better-correlated outputs but can also quantify the uncertainty estimates.

### 4.3 Results

Table 1 shows the performance of various methods on different tasks and datasets. For all the regression tasks, we compute MSE and MAE to measure the discrepancy between the predictions and the groundtruth. Moreover, Spearman and Pearson correlation coefficients indicate the positive association between the predicted point estimates from the network and the groundtruth target variable. A high correlation indicates that the prediction will be able to indicate the severity of the medical condition (which may help prescribe the treatment as explained in Section 1). For probabilistic models, the quality of uncertainty estimates is measured by UCE and NLL (Laves et al., 2020a; Upadhyay et al., 2022). In addition, we also quantify the calibration of the estimated uncertainty using the interval that covers the true value with a probability of $\alpha$ (under the assumption that the predicted distribution is correct) is given by,

$$\mathcal{I}_\alpha(\hat{\mathbf{y}}, \hat{\sigma}) = \left[\hat{\mathbf{y}} - \frac{C^{-1}(\alpha)}{2}\hat{\sigma}, \hat{\mathbf{y}} + \frac{C^{-1}(\alpha)}{2}\hat{\sigma}\right] \tag{11}$$

Where, $C^{-1}(\cdot)$ refers to the quantile function, i.e., inverse of the cumulative distribution function, discussed thoroughly in (Kuleshov et al., 2018; Gawlikowski et al., 2021). In particular, we set the $\alpha = 0.95$, which makes $C^{-1}(0.95) = 1.645$ for Gaussian distribution, and the length of the interval $\mathcal{I}_{0.95}(\hat{\mathbf{y}}, \hat{\sigma})$ becomes, $len(\mathcal{I}_{0.95}(\hat{\mathbf{y}}, \hat{\sigma})) = C^{-1}(0.95)\hat{\sigma} = 1.645\hat{\sigma}$. For good calibration, $\alpha$ fraction of groundtruth samples should lie in the interval $\mathcal{I}_\alpha(\hat{\mathbf{y}}, \hat{\sigma})$ and the length of the interval, $C^{-1}(\alpha)\hat{\sigma}$, should be small. We report both metrics in our evaluation. For the interval length, we report the mean interval length of the test set, represented by $< len(\mathcal{I}_\alpha(\hat{\mathbf{y}}, \hat{\sigma})) >$. Finally, as discussed in Section 3.3.3, regression model also allows classification, which is the approach taken by some of the prior works. To compare them, for the binary classification tasks, we also report the AUC, Sensitivity, Specificity, and Positive/Negative Predictive values (PPV/NPV).

#### 4.3.1 Task Specific Performance

As indicated in Table 1, for *Survival Estimation*, the regression model trained with L2 loss (`Regres.-L2`) has a MSE of 100.12 (MAE of 8.49, Spearman/Pearson correlation of 0.30/0.30), which are worse than the performance of the model trained with L1 loss (`Regres.-L1`) (MSE of 73.21, MAE of 7.78, Spearman/Pearson correlation of 0.38/0.41), this is in line with previous works. Moreover, we could further improve the performance of the regression model trained with L1 loss by incorporating kernel density-based weights (`Regres.-L1-KDEw`), as described in Section 3.3, that accounts for imbalance in the dataset and leads to a performance with MSE of 67.97 (MAE of 6.30, Spearman/Pearson correlation of 0.51/0.53). Additionally, we notice that heteroscedastic Gaussian probabilistic regression model (`Regres.-w.-U`) from (Von Bachmann et al., 2022) performs better than homoscedastic Gaussian model (`Regres.-L2`) but not as good as (`Regres.-L1`). However, `Regres.-w.-U` estimates the uncertainty in the prediction (with UCE/NLL

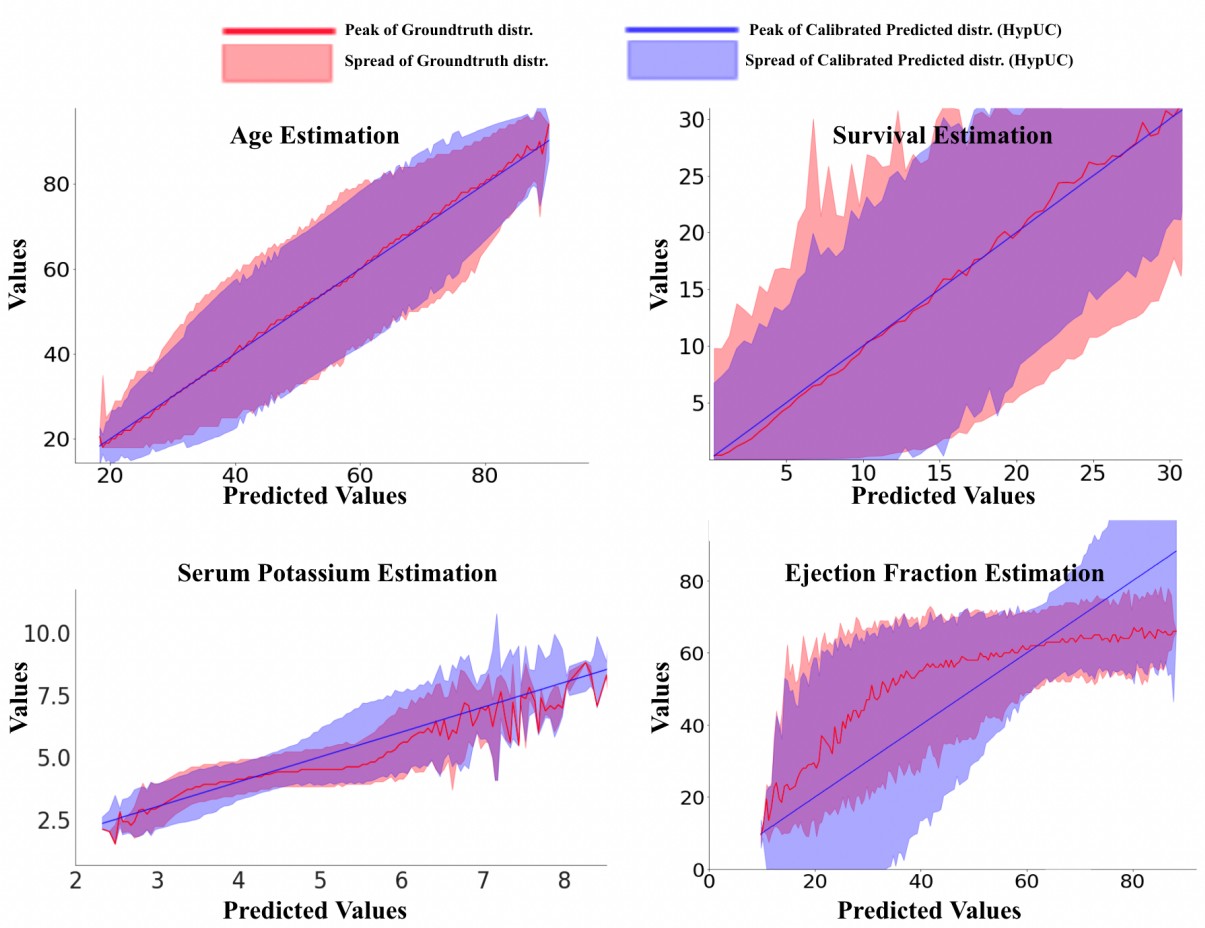

**Figure 7:** Groundtruth and predicted distributions for 4 different tasks (Age/Survival/Serum Potassium/LVEF estimation). Predicted distribution from `HypUC` tightly encompasses groundtruth distribution (compared to predicted distribution from `Regres.-w.-U`) indicating predictions are well-calibrated.

of 2.37/5.89), but our proposed `HypUC` not only improves regression performance (with MSE/MAE of 54.62/5.38) but the quality of uncertainty as well (with UCE/NLL of 0.58/3.45). We observe a similar trend for the *Age Estimation*.

For *Serum Potassium Estimation*, we compared with a binary classifier baseline (`Binary-Classifier` in Table 1), an approach found in (Galloway et al., 2019), which predicts if a given ECG corresponds to Hyperkalemia or not, instead of estimating the serum potassium level. While such a binary classifier achieves a good classification performance as indicated by an AUC of 0.87, it does not gauge the severity of the condition that may decide the diagnosis. Figure 6 shows the correlation between the ground truth serum potassium level compared to the scores derived from binary classifier (Figure 6-left and middle), the lack of high positive correlation (Spearman correlation of 0.39) indicates that scores from the model can not be used as an indication for the severity of the medical condition (within Hyperkalemia), but can be used to discriminate Hyperkalemia from non-Hyperkalemia cases. We also train different kinds of regression models to regress the serum potassium level. As indicated in Table 1, we see that regression models generally produce outputs that are better correlated with the groundtruth (e.g., `Regres.L2` has a Spearman correlation coefficient of 0.51 vs. 0.39 for `Binary Classifier`). Moreover, we observe that among deterministic regression models `Regres.-L1-KDEw` performs the best with MSE of 0.19 (MAE of 0.33). The probabilistic regressor `Regres.-w.-U` performs better than `Regres.-L2` and provides uncertainty estimates as well (UCE/NLL of 1.83/8.76). But, `HypUC` provides the best regression performance along with better uncertainty estimates. Additionally, the outputs from the regression models can be used to classify the ECG into Hyperkalemia or

not (similar to binary classifier), we notice that all the regression models perform comparable classification to classifier. In particular, `HypUC` has AUC of 0.89 which is higher than the binary classifier (AUC of 0.87).

A similar trend is observed for *Left Ventricular Ejection Fraction (LVEF) Estimation* where the proposed `HypUC` performs well in regression along with better-calibrated uncertainty estimates. Moreover, when used as a classifier, it also yields a better performance than the binary classifier trained from scratch, emphasizing the benefits of tackling this problem with the proposed probabilistic regression framework, `HypUC`.

### 4.3.2 Calibration of Uncertainty Estimates on the Test-set

Figure 7 shows the groundtruth and the predicted distribution for Age/Survival/Serum Potassium/LVEF estimation tasks on the test dataset. The x-axis in the plot indicates the predicted point estimates from the `HypUC`. For a given point on x-axis: (i) Red bold curve shows the peak of the groundtruth distribution (i.e., the distribution of groundtruth values that correspond to predicted value given by the x-axis). (ii) Shaded red region indicate the spread of the groundtruth distribution (i.e., 2 and 98 percentile of the groundtruth distribution). (iii) Blue bold curve shows the peak of the `HypUC`-predicted distribution. (iv) Blue shaded region indicates the spread of the groundtruth distribution (i.e., 2 and 98 percentile of the groundtruth distribution). We notice that predicted distribution from `HypUC` tightly encompasses groundtruth distribution (compared to predicted distribution from `Regres.-w.-U`) indicating predictions are well-calibrated. We calibrate `HypUC` using small validation sets but it generalizes well to much larger test sets.

### 4.3.3 Entropy-based Filtering to Trigger Human Expert Intervention

As the proposed `HypUC` produces well-calibrated predictive distribution, we propose a technique to use the predicted distribution to flag the unreliable predictions. For a given input sample $\mathbf{x}$, the trained DNN produces $\{\hat{\mathbf{y}}, \hat{\sigma}\} = \mathbf{\Psi}(\mathbf{x}; \theta^*)$. We then apply the post-hoc calibration techniques to obtain $\{\hat{\mathbf{y}}, \hat{\sigma}_{calib}\}$. We then compute the entropy of the predicted distribution, i.e., $\mathcal{N}(\hat{\mathbf{y}}, \hat{\sigma}^2_{calib})$ given by $\mathcal{H}(\mathbf{z})$, i.e.,

$$\mathcal{H}(\mathbf{z}) = \int -p(\mathbf{z})\log p(\mathbf{z})d\mathbf{z} \text{ , where } \mathbf{z} \sim \mathcal{N}(\hat{\mathbf{y}}, \hat{\sigma}^2_{calib}) \tag{12}$$

$$\mathcal{H}(\mathbf{z}) = -\mathbb{E}(\log \mathcal{N}(\hat{\mathbf{y}}, \hat{\sigma}^2_{calib})) = \log \sqrt{2\pi e \hat{\sigma}^2_{calib}} \tag{13}$$

At the test time, if for the sample $\mathbf{x}$ the entropy $\log \sqrt{2\pi e \hat{\sigma}^2_{calib}} > \tau_{q,val}$ then the prediction is flagged as unreliable. The threshold $\tau_{q,val}$ is computed using the validation set as the $q-\%$tile value from the distribution of entropy values as shown in Figure 8 for LVEF estimation using `HypUC`. Figure 9 evaluates the regression performance of `HypUC` (in terms of MAE) on the test sets for various tasks by removing the unreliable predictions based on different thresholds derived from the validation set. We observe that as we remove the higher fraction of unreliable samples (i.e., decreasing the threshold $\tau_{q,val}$ by lowering the value of $q$) the performance improves. This phenomenon demonstrates that we could enhance the predictions made by the model by only showing reliable predictions and flagging the unreliable predictions (based on entropy measure) to be evaluated by human experts which may be critical for deploying machine learning models for healthcare applications in the real world.

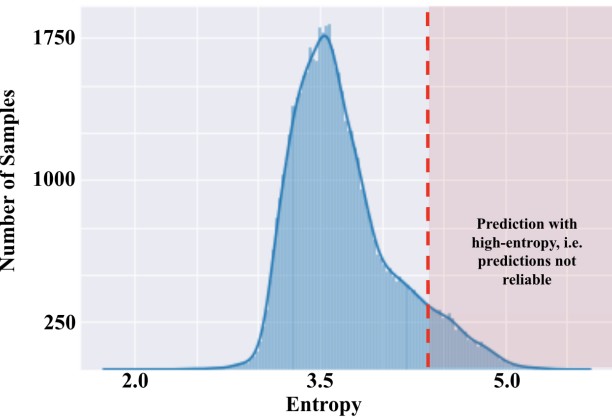

**Figure 8:** Distribution of entropy for LVEF validation set with a threshold of 10%-tile.

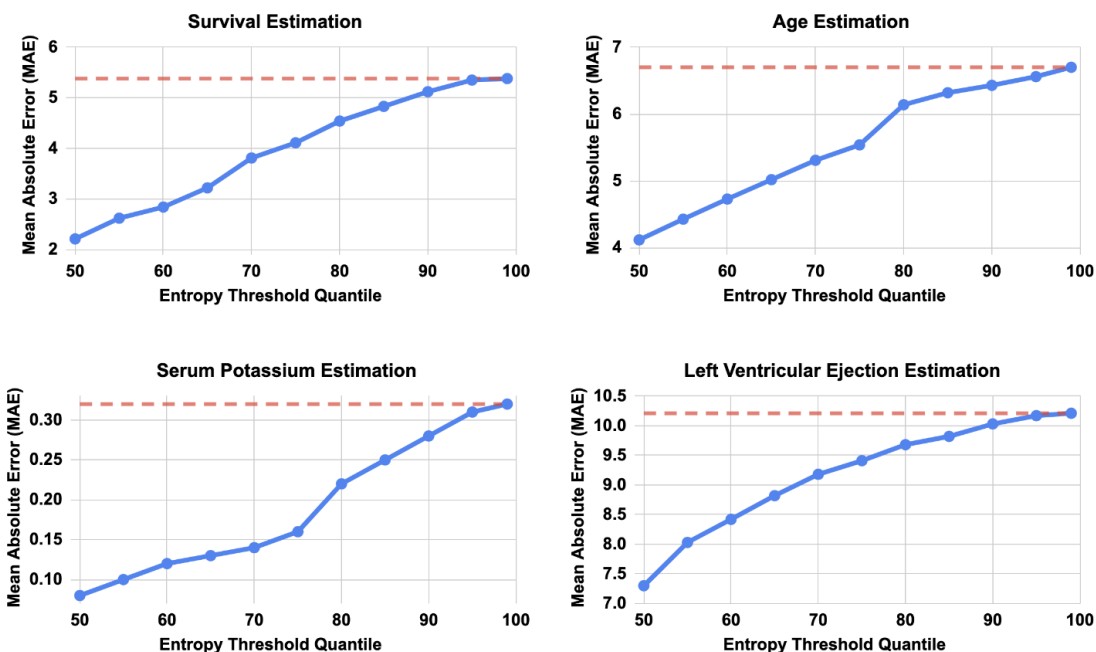

**Figure 9:** Evaluating `HypUC` on test-set by removing the unreliable (i.e., high entropy) predictions. We see consistent improvement as we decrease the entropy threshold based on quantile as described in Section 4.3.3

## 5 Discussion, Conclusion, and Future Work

Automated analysis of medical time series (e.g., ECG) using ML techniques opens several avenues in large-scale remote healthcare assistance while making more efficient use of existing procedures. This work proposes `HypUC` to solve the imbalanced regression problems with medical time series along with uncertainty estimation. We also introduced a new hyperfine uncertainty calibration technique that provides reliable uncertainty estimates for a diverse range of test sets. Additionally, we present a method for using these calibrated uncertainties to improve decision-making through an ensemble of gradient-boosted learners. In our studies, we use the well-calibrated predictive distribution derived from `HypUC` in an entropy-based technique to flag the unreliable predictions at the inference time for human expert evaluation and improve the performance of the ML model. Our approach is demonstrated on a large, real-world dataset of ECGs from millions of patients with various medical conditions, and is shown to outperform several baselines while also providing calibrated uncertainty estimates. Note that, while `HypUC` presents a technical solution to the above-mentioned problems, it is crucial to explore the clinical impact of our work using real-world validation studies. Additionally, it is important to extend our method to predict multiple target values in a single forward pass (i.e., multi-task learning) to avoid training models for different diseases and allow cross-task learning. Another important aspect for clinical deployment is to design protocols that will help physicians make decisions in the presence of quantified uncertainty, which is much needed. These challenges are going to be explored in a near future clinical study for hyperkalemia detection algorithm based on our proposed method.

**Broader Impact.** The large-scale real-world datasets from millions of patients were obtained from `Mayo Clinic, USA` following the guidelines for retrospective study with *deidentified data* approved by the institutional review board of `Mayo Clinic, USA` following the state-of-the-art identification protocol as described in Murugadoss et al. (2021). While this study is done on a large real-world anonymized dataset, the ML models trained in this study must be evaluated for bias. Typically, formal clinical deployment of this technique will require validation and testing on demographically diverse datasets.

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
