# OpenReview forum: "HypUC: Hyperfine Uncertainty Calibration with Gradient- boosted Corrections for Reliable Regression on Imbalanced Electrocardiograms"
_TMLR — Accepted by TMLR_

### Review · Reviewer_DNgP · 2023-05-26

**Summary Of Contributions:**

The paper presents several contributions and new knowledge in the field of medical time series analysis. Firstly, it addresses the challenge of imbalanced regression in medical time series by proposing the HypUC framework. This framework incorporates probabilistic machine learning techniques and a hyperfine uncertainty calibration method to accurately predict continuous-valued physiological parameters. By focusing on regression rather than classification, the paper expands the scope of previous research in the field.

Secondly, the paper introduces a novel approach to uncertainty estimation in medical time series analysis. The calibrated uncertainties derived from the HypUC framework provide reliable estimates that generalize well to diverse test sets. This is crucial for clinical decision-making and enables clinicians to assess the confidence and reliability of the predictions.

Furthermore, the paper presents an ensemble approach that utilizes the calibrated uncertainties to improve decision-making. By combining an ensemble of gradient-boosted learners, the paper demonstrates how the calibrated uncertainties can enhance the performance of the machine learning model. This ensemble approach contributes to more accurate and reliable predictions, leading to improved clinical outcomes.

Additionally, the paper introduces an entropy-based technique to flag unreliable predictions. This technique helps identify cases where the model may not be providing trustworthy predictions, enabling human expert evaluation and intervention when necessary. This contributes to the overall reliability and interpretability of the proposed approach.

The evaluation of the HypUC framework on a large, real-world dataset of ECGs collected from millions of patients with various medical conditions demonstrates its superior performance compared to several baselines. The framework achieves accurate predictions and provides calibrated uncertainty estimates, making it suitable for clinical use and real-world deployment.

In summary, the contributions of this paper include the development of the HypUC framework for imbalanced probabilistic regression in medical time series. The introduction of reliable uncertainty estimation, the utilization of calibrated uncertainties to improve decision-making, and the application of an entropy-based technique for identifying unreliable predictions are important contributions to the field. These contributions advance the field of medical time series analysis and have the potential to enhance diagnostic decision-making and remote healthcare assistance.

**Audience:**

Yes

**Broader Impact Concerns:**

While the paper provides valuable contributions to the field of automated analysis of medical time series, there are a few ethical implications that should be addressed in the Broader Impact Statement. Specifically:

- Privacy and Data Security: Given the use of large-scale datasets collected from millions of patients, it is crucial to address privacy concerns and ensure appropriate data protection measures are in place. The paper should discuss how patient privacy was protected during data collection, storage, and analysis, as well as any steps taken to anonymize or de-identify the data.

- Bias and Fairness: The use of machine learning algorithms in healthcare raises concerns about bias and fairness. The authors should acknowledge the potential for bias in the dataset used and discuss steps taken to mitigate and monitor bias in the algorithmic predictions. Additionally, the paper should address the fairness considerations in terms of the impact on different demographic groups or subpopulations.

- Clinical Validation and Human-in-the-Loop: While the proposed approach shows promising results, it is important to emphasize the need for clinical validation and the importance of human experts in the decision-making process. The paper should discuss how the proposed framework can be integrated into clinical workflows and emphasize the collaborative role of healthcare professionals in interpreting and utilizing the algorithmic predictions.

- Deployment Challenges and Societal Impact: The authors should address potential challenges and barriers to the real-world deployment of the proposed framework. This includes considerations of cost, infrastructure requirements, acceptance by healthcare providers, and the potential impact on healthcare disparities. Discussing these aspects would provide a more comprehensive understanding of the broader impact of the work.

By addressing these ethical concerns in the Broader Impact Statement, the authors can ensure a more comprehensive assessment of the potential implications and contribute to a responsible and ethically sound implementation of the proposed framework.

**Claims And Evidence:**

Yes

**Requested Changes:**

Proposed Changes:

- Strengthen the Related Work section by including recent work in the healthcare domain and other related domains (doi.org/10.1145/3313831.3376506, doi.org/10.1145/3132272.3134111). This would provide a stronger basis for the proposed approach and demonstrate the novelty of the contribution.

- Provide a more detailed explanation of the methodology and implementation details (doi.org/10.1109/JBHI.2023.3271615, doi.org/10.1016/j.ijhcs.2022.102922). Specifically, provide a step-by-step overview of the HypUC framework and the ensemble approach. This would help readers understand the proposed methods more clearly.

- Include a more in-depth analysis and discussion of the results, particularly insights into the performance of the proposed approach on different medical conditions or subgroups of patients (doi.org/10.1007/s10877-022-00873-7, doi.org/10.1145/3544548.3580682). This would enhance the paper's impact and applicability.

- Discuss potential limitations and challenges associated with the proposed approach. Addressing these limitations and outlining future directions would strengthen the paper.

- Improve the organization and structure of the paper to ensure a clear flow of ideas and logical progression.

- Address proofreading and formatting issues to enhance the overall quality and readability of the manuscript.

These proposed changes, while not critical to securing a recommendation for acceptance, would significantly strengthen the work by addressing important aspects and improving clarity, comprehensiveness, and impact.

**Strengths And Weaknesses:**

Overall, the paper showcases significant strengths in addressing an important problem, proposing a novel framework, and demonstrating its effectiveness. Attention to the weaker elements mentioned would further improve the clarity, comprehensiveness, and impact of the manuscript.

1. Strengths:

1.1. The paper addresses a significant and practical concern in the field of medical time series analysis, specifically the regression of continuous-valued physiological parameters.

1.2. The suggested HypUC framework incorporates probabilistic machine learning techniques and a hyperfine uncertainty calibration method, providing reliable uncertainty estimates in imbalanced regression tasks.
The ensemble approach, utilizing calibrated uncertainties, improves decision-making and enhances the performance of the machine learning model.

1.3. The paper demonstrates the effectiveness of the proposed approach on a large, real-world dataset of ECGs, showcasing its superior performance compared to several baselines.
The entropy-based technique for flagging unreliable predictions adds an critical layer of interpretability and trustworthiness to the proposed approach.

1.4. The paper provides a comprehensive overview of the proposed framework, its implementation, and evaluation, allowing for reproducibility and further research.

2. Weaknesses:

2.1. The related work section could be more comprehensive, discussing recent work in the healthcare domain and other related domains. This would provide a stronger basis for the proposed approach and demonstrate the novelty of the contribution.

2.2. The paper would benefit from a more detailed explanation of the methodology and implementation details. Providing a step-by-step overview of the HypUC framework and the ensemble approach would help readers understand the proposed methods more clearly.

2.3. While the evaluation on a large real-world dataset is commendable, more detailed analysis and discussion of the results could be included. Specifically, providing insights into the performance of the proposed approach on different medical conditions or subgroups of patients would further strengthen the paper.

2.4. The paper could consider discussing potential limitations and challenges associated with the proposed approach. Addressing these limitations and outlining future directions would enhance the impact and applicability of the research.

---

> ### Author Response · Authors · 2023-06-11
> **Response to reviewer DNgP**
>
> We sincerely thank the reviewer for providing a detailed and thoughtful feedback. We are glad to know that review acknowledged our work with several contributions and new knowledge in the field of medical time series analysis by addressing the challenge of imbalanced regression in clinical setup with probabilistic machine learning techniques and a hyperfine uncertainty calibration method to accurately predict continuous-valued physiological parameters.
>
> Indeed our focus on regression rather than classification, was motivated by the possible expansion of the scope of previous ML research in this direction as rightly pointed out by the review.
>
> We have revised the manuscript (revised text in blue) to accommodate the suggestions in the review and address the queries below:
>
> **Refining the flow, details of the method, and related work**
>
> * We have updated Section-1,3,4,5 to better structure the flow of the paper along with highlighting the details of the various steps of our method, HypUC, and how the various design choices address some of the limitations in the existing literature
>
> * We've also updated the related work section to clearly highlight the previous work in this domain and their limitations along with the solutions offered by our proposed method. We also thank the reviewer for pointing us to relevant related works, we've included the same in the manuscript.
>
> **More analysis and discussing limitations**
>
> *  We've incorporated new experiments, results and analysis in section-4 discussing more possible approaches to handle the imbalanced regression dataset. Indeed, cohort-based analysis for different medical subgroups will definitely be beneficial and we are working on a similar setup to allow clinical review of the same. However, such an analysis require extensive interaction with medical practitioners as well as regulatory board for designing of such cohort, releasing the data, verifying the details and results of the experiments, which is beyond the scope of this work, and perhaps better suited for a clinical study, which as noted in the revised manuscript is set to start in near future.
>
> * As suggested in the review, we have now included a discussion on limitation of our work and possible future work directions that may further enhance the capabilities in Section-5. Among these, are efforts to build a multi-head version of our method to allow simultaneous prediction of multiple physiological parameters along with the uncertainty estimates. This would avoid training models for different diseases and also allow cross-task learning. Moreover, elements to make our models more robust are also desirable. Another important aspect for clinical deployment is also to design protocols that help physicians in decision making, even in the presence of quantified uncertainty.
>
> **Addressing broad impact concerns**
>
> * As rightly pointed out in the review, it is important to honour data security and privacy regulations. We have updated the manuscript to discuss the data acquisition and usage guideline set by the institutional review board of DEIDENTIFIED-MEDICAL-SITE.
>
> * We also note in Section-5 that while this study is done on a large real-world anonymised dataset, the ML models trained in
> this study are likely biased because of the demography of patients available at DEIDENTIFIED-MEDICAL-SITE. Clinical deployment of this technique to different sites will call for diverse dataset collection and improving the modeling aspect to make it robust and unbiased.
>
> * Moreover we also note that, real-world deployment to various parts of the world would also require successful clinical trials to be approved by regulatory institutions.

---

> > ### Comment · Reviewer_DNgP · 2023-06-11
> > **Approval of Manuscript Revisions and Acknowledgment of Enhanced Contributions to Medical Time Series Analysis**
> >
> > Thank you for your comprehensive response to my comments. I appreciate your effort in addressing the feedback and amending your manuscript accordingly. The changes made in the manuscript, especially those relating to the flow and detailing of the method, the expansion of the related work section, the additional analyses, and the discussion on limitations, indeed enhance the paper significantly.
> >
> > Your plans for future work are particularly intriguing, such as the development of a multi-head version of your method and designing protocols to assist physicians in decision-making, even in the face of quantified uncertainty.
> >
> > Moreover, the inclusion of a section that addresses broad impact concerns is greatly appreciated. It shows a thoughtful and responsible approach to the development and potential deployment of your technique.
> >
> > In conclusion, I am satisfied with the revisions made and believe that the paper has been significantly improved as a result. This work is an important contribution to the field of medical time series analysis.

---

> > > ### Author Response · Authors · 2023-06-13
> > > **Thank you for your contributions!**
> > >
> > > Respected reviewer,
> > >
> > > Thank you for your comments and contributions!

---

### Review · Reviewer_3msx · 2023-05-29

**Summary Of Contributions:**

This paper studies regression tasks on physiological signals, specifically focusing on using ECG to diagnose cardiovascular diseases. The proposed approach combines kernel density estimation and uncertainty estimation techniques.

**Audience:**

No

**Broader Impact Concerns:**

One sentence in the abstract (also conclusion) says that the proposed approach "is suitable for clinical use and real-world deployment". This is a very strong claim that necessitates further discussion and justification. For example, how are medical professionals performing the task that the ML model is solving, but without ML? How well do they perform currently? How important or useful are tasks such as "estimating age from ECG"? Without addressing these, I disagree with the statement that the proposed approach is ready for real-world clinical use.

**Claims And Evidence:**

No

**Requested Changes:**

- Please define the notation before using them. For example, on page 4, Sec 3.2.3 Uncertainty Estimation, the symbols $\zeta$ and $\hat{\nu}_i$, $\hat{\rho}_i$ were used without being defined. I'm guessing $\zeta$ is the parameters of the DNN, but I still have no idea what $\\{\hat{\boldsymbol{y}}_i, \hat{\nu}_i, \dots, \hat{\rho}_i\\}$ refers to. This has greatly hindered my understanding of the paper.
- Please update the related work section to explain how each past work is related and how the present work differs. Right now, the realted work section simply lists prior work in the format of "A study.... Another study...There have been studies" without explicitly saying what previous work hasn't achieved and what this work adds. Readers not too familiar with the field (such as myself) will find it tempting to skip this section as it reads like an enumeration of papers about deep learning on ECG, but there's no discussion about why or how they are related specifically to this work.
- Please update in-text citations to distinguish between \citet vs \citep

**Strengths And Weaknesses:**

- Unfortunately I found this submission quite difficult to follow. The mathematical notation are not clearly introduced and it appears unclear to me how the "hyperfine uncertainty calibration" step and the "ensemble of gradient-boosted learners" step are central to the proposed approach.
- I'm not sure if TMLR is a suitable venue for this submission. Based on the [editorial policies](https://jmlr.org/tmlr/editorial-policies.html), it doesn't seem to fit under any of the items under "Scope". Perhaps the closest one is "accounts of applications of existing techniques that shed light on the strengths and weaknesses of the methods"; however, I believe that this submission is still lacking in these aspects as it does not seem to "shed light on the strengths and weaknesses of the methods".

---

> ### Author Response · Authors · 2023-06-11
> **Response to reviewer 3msx - part 1**
>
> ***(This is part 1, the next comment is part 2)***
>
> We thank the reviewer for the comments. We address the queries below
>
> **Mathematical notation**
>
> * We revamped the manuscript to make sure that the mathematical notations are well defined.
> * Note that Section-3.2.3 is preliminary on uncertainty estimation, and describes the popular approach to quantify the uncertainty estimates as thoroughly discussed in many previous works. We explicitly clarify that the model must estimate the parameters of the conditional probability distribution, $\mathcal{P}_{Y|X}$,
> which is used to maximize the likelihood. This distribution is assumed to be parametrized by set of parameters $\\{\hat{\mathbf{y}}, \hat{\nu}_1, \hat{\nu}_2 \dots \hat{\nu}_k\\}$ which we need to estimate (e.g., for Laplace distribution the set of parameters is given by location and scale, $\\{\mu, b\\}$, for Gaussian distribution the set of parameters is given by mean and standard-deviation, i.e, $\\{\mu, \sigma\\}$, for generalized Gaussian the set of parameters is given by location, scale, and shape parameters $\\{\mu, \alpha, \beta\\}$.
> * Different choice of distribution will require different set of parameters, so we represent the arbitrary parameterized $\mathcal{P}_{Y|X}$ distribution by a set of parameters $\\{\hat{\mathbf{y}}, \hat{\nu}_1, \hat{\nu}_2 \dots \hat{\nu}_k\\}$.
>
> **Hyperfine uncertainty calibration**
>
> * Note that our proposed Hyperfine uncertainty calibration is crucial step because as clarified in the Section-3.2.3, the existing approach do not yield well-calibrated uncertainty estimates, hindering its applications in critical scenarios.
> * While previous works perform post-hoc calibration (i.e., post training) by scaling the uncertainty, these are based on single scaling factor that is learned with the entire dataset -- this approach does not work well with highly imbalanced datasets (e.g., in clinical setups).
> * Our proposed hyperfine calibration step provides a first solution for this problem. Moreover, section-4.3.2, 4.3.3 also demonstrate the efficacy of well-calibrated uncertainties obtained by our method in practical setup.
>
> **Gradient-boosted learners**
> * While the previous part of our method proposes how to estimate well calibrated uncertainties, this part highlights the use of well-calibrated uncertainty in practical setup.
> * Gradient boosted learners based on the predictions made by DNN are crucial component that shows well-calibrated uncertainties can further improve the quality of predictions for decision making as discussed in section-3.3.3
> * Our work provides first set of empirical results on how well-calibrated uncertainty estimates in this domain (i.e., healthcare time-series) can potentially help in the decision making.
>
> **Relevance to TMLR**
>
> * This work makes several *technical contributions in machine learning* including:
>
>   * KDE based reweighing scheme to tackle imbalanced regression combined with probabilistic regression to allow uncertainty estimation, discussed in Section-3.2, 3.3
>   * A new approach for calibrating the predicted uncertainty, discussed in Section-3.3.2.
>   * We demonstrate the efficacy of the calibrated uncertainty estimates to improve the predicted continuous value and also flag unreliable predictions, as discussed in Section-3.3.3
>
> *(The above is consistent with observations made in other reviews)*
>
> * We refer to [TMLR sub. guideline](https://www.jmlr.org/tmlr/editorial-policies.html), and highlight that due to above contributions we fit well within the scope as defined by (i) new algorithms with sound empirical validation; (ii) experimental studies yielding new insight into the design and behavior of learning in intelligent systems; (iii) formalization of new learning tasks (e.g., in the context of new applications) and of methods for assessing performance on those tasks; (iv) new approaches for analysis, and understanding of AI/ML systems.
>
> * Moreover, we highlight several TMLR accepted work that are making ML contributions using healthcare/medical domain as an application example:
>   * [PCPs: Patient Cardiac Prototypes to Probe AI-based Medical Diagnoses, Distill Datasets, and Retrieve Patients](https://openreview.net/pdf?id=X1pjWMCMB0)
>   * [Failure Detection in Medical Image Classification: A Reality Check and Benchmarking Testbed](https://openreview.net/pdf?id=VBHuLfnOMf)
>   * [Differentially Private Fréchet Mean on the Manifold of Symmetric Positive Definite (SPD) Matrices with log-Euclidean
> Metric](https://openreview.net/pdf?id=mAx8QqZ14f)
>   * [Conformal Prediction Intervals with Temporal Dependence](https://openreview.net/pdf?id=8QoxXTDcsH)
>   * [On the Choice of Interpolation Scheme for Neural CDEs](https://openreview.net/pdf?id=caRBFhxXIG)
>   * [Beyond Information Gain: An Empirical Benchmark for LowSwitching-Cost Reinforcement Learning](https://openreview.net/pdf?id=Xq1sTZTQVm)

---

> ### Author Response · Authors · 2023-06-11
> **Response to reviewer 3msx - part 2**
>
> ***(This is part 2, to be read after part 1 in another comment)***
>
> **Related work, citations, and broader impact**
>
> * We have revamped our related work section to highlight what each one of them do and what is lacking in all the existing approaches, and the fix offered by our work. We've also carefully fixed the `\cite` vs `\citep` in the manuscript.
>
> * We have also clarified that while our work proposes a technical solution for a relevant problem in healthcare. The points made by the reviewer are indeed valid and it is important to conduct prospective and clinical validation. Indeed, we have been engaged in a follow up work with clinical medical researchers and note that prospective clinical trial based on our algorithm is now sponsored, sanctioned, and set to ensue in the near future. We have duly noted this in our revised manuscript (in Abstract, Section-5, etc, in blue).
>
> * Estimating cardiovascular age from ECG is an actively studied area as it acts as indicator of overall cardiovascular health, some examples are:
>   * [The effect of age on the electrocardiogram](https://www.sciencedirect.com/science/article/abs/pii/0002914972904171)
>   * [Changes in ECG pattern with advancing age](https://www.degruyter.com/document/doi/10.1515/JBCPP.2011.017/html)
>   * [Cardiac age detected by machine learning applied to the surface ECG of healthy subjects: Creation of a benchmark](https://www.sciencedirect.com/science/article/abs/pii/S0022073622000395)
>   * [Artificial Intelligence–Derived Electrocardiogram Assessment of Cardiac Age and Molecular Markers of Senescence in Heart Failure](https://www.sciencedirect.com/science/article/abs/pii/S002561962200619X)
>   * [Can functional cardiac age be predicted from the ECG in a normal healthy population?](https://ieeexplore.ieee.org/abstract/document/6420340)
>   * [Deep learning-derived cardiovascular age shares a genetic basis with other cardiac phenotypes](https://www.nature.com/articles/s41598-022-27254-z)

---

> ### Comment · Reviewer_3msx · 2023-06-15
>
> Dear authors, thank you for updating the manuscript. With the edits, the manuscript has become clearer and improved my understanding of this work. The updated method section is much clearer compared to the initial version. Many of the claims have also been adjusted to better reflect what can be supported by experimental evidence.
>
> After reading the updated manuscript, I have the following feedback:
> - Table 1 results: for the main metrics and comparisons with baselines, can you provide error bars and test whether the differences are statistically significant?
> - Table 1 Spearman/Pearson: for age estimation the correlation is about 0.8 which is very good, but for the other three tasks the correlation is much lower even with the proposed approach HypUC (about 0.4-0.6). I also appreciate that HypUC outperforms the baselines which attained correlation of about 0.3 on these tasks. Do you have any insights for why certain tasks might be easier or harder? Is a correlation of 0.5 useful?
> - Fig 7: for age, survival, and serum potassium, the predicted distributions line up quite well ground-truth in each case. But for LVEF, the predicted distribution is wider than the ground-truth, and it has an interesting shape with a "narrowing" around the value 60. Do you have any insights for why this might be the case?

---

> > ### Author Response · Authors · 2023-06-24
> > **Response to reviewer 3msx**
> >
> > We thank the reviewer for acknowledging that the updates in the manuscript has made things clear.
> > We address the queries below:
> >
> > **Statistical significance**
> >
> > As mentioned in the review, this is certainly an important point. We wanted to emphasise that:
> > - Even with a similar regression performance for a deterministic regression model and a probabilistic regression model, the probabilistic model allows for uncertainty quantification at the sample level that is not possible with a deterministic regression model.
> > - For the proposed HypUC model (which is a probabilistic regression model), it needs to significantly improve over *other probabilistic regression* models both in terms of regression performance and in terms of uncertainty calibration performance.
> > - We note that while comparing HypUC (proposed probabilistic regression framework) with Regres.-w.-U (baseline probabilistic regression framework):
> >     - (i) regression performance is significantly better for HypUC, i.e.,
> >
> >              [Survival estimation] MSE of 54.62 (2.6) vs. 84.62 (3.3) with p < 0.05
> >              [Age estimation] MSE of 74.9 (4.4)  vs. 136.27 (8.2) with p < 0.05
> >              [Serum Potassium estimation] MSE of 0.20 (0.03) vs. 0.23 (0.06) with p < 0.05
> >              [LVEF estimation] MSE of 133.26 (10.4) vs. 165.81(13.6) with p < 0.05
> >
> >     - (ii) the uncertainty calibration performance is also significantly better for HypUC, i.e.,
> >
> >              [Survival estimation] UCE of 0.58 (0.04) vs. 2.37 (0.83) with p < 0.05
> >              [Age estimation] UCE of 1.06 (0.11)  vs. 13.32 (0.64) with p < 0.05
> >              [Serum Potassium estimation] UCE of 0.41 (0.06) vs. 1.83 (0.16) with p < 0.05
> >              [LVEF estimation] UCE of 1.68 (0.14) vs. 10.56 (0.62) with p < 0.05
> >
> >
> > **Difficulty of tasks and correlation coefficient**
> >
> > - As observed in the review, indeed, for certain tasks (e.g., Age estimation) the performance of all the models are overall better than other tasks (e.g., LVEF). This majorly indicates the relative skew of the label distribution for the available datasets for different tasks and also the presence of necessary signatures in the time-series to extract information about those tasks. For instance, the changes in ECG with cardiovascular age are more prominent.
> > - It is important to note that higher correlation coefficient is desirable, relative to a lower correlation coefficient (e.g., for Task A a method with correlation of 0.4 is better than the method with correlation of 0.2). However, a single correlation coefficient in isolation does not convey much information (e.g., for Task A the method has a correlation coefficient is 0.5 -- this alone does not fully communicate if the method is useful for that task or not). One has to rely on multiple metrics and also gauge relative performances (w.r.t other methods). As seen in Table 1, HypUC consistently provides good regression as well as uncertainty calibration performance as indicated by collective set of metrics across tasks.
> >
> > **LVEF distribution**
> >
> > - We thank the reviewer for highlighting the observation regarding narrowing of the LVEF distribution. Our investigation shows that one potential reason for this phenomena can be attributed to the fact that:
> >   - The normal range for ejection fraction is in the range 50% - 60% [a,b]
> >   - The disease conditioned that is often diagnosed is the low LVEF range, i.e., <40% [a,b]
> >   - The presence of the hyper-dynamic LVEF range, i.e., >70% is very rare and have much lower prevalence
> >
> >
> > **Reference**
> >
> > [a] [Left Ventricular Ejection Fraction](https://www.ncbi.nlm.nih.gov/books/NBK459131/#:~:text=Hyperdynamic%20%3D%20LVEF%20greater%20than%2070,to%2039%25%20(midpoint%2035%25))
> >
> > [b] [Heart Left Ventricle Ejection Fraction](https://www.sciencedirect.com/topics/nursing-and-health-professions/heart-left-ventricle-ejection-fraction)

---

### Review · Reviewer_oWh4 · 2023-05-29

**Summary Of Contributions:**

The paper address the imbalanced label problem in medical time series problem. The model first uses a weighted loss function based on KDE to learn more on the data with minority labels. Then it recalibrates the prediction with a post-processing step. The proposed method outperforms the vanilla regression methods.

**Audience:**

Yes

**Broader Impact Concerns:**

No.

**Claims And Evidence:**

Yes

**Requested Changes:**

- Clarify what the main purpose of this paper is.  Modifying and restructuring some sections that have very diverse arguments can help the reader understand the paper better.

- Conduct the suggested experiment. Choose the correct calibration metric if it is one of the contributions in the paper.

**Strengths And Weaknesses:**

Strength:

- It is quite interesting to deal with the imbalanced data in the regression problem by using KDE to reweight the loss as analogous to using class weights to reweight the loss in the classification problem.

- The method has significant improvements on the baseline methods in multiple regression tasks on the ECG data.


Weakness:

- Need to compare with some simple baselines that deal with long-tail regression problems. For instance, taking logarithms or standardizing the label during training.

- The evaluation on calibration is insufficient. If the authors would like to claim that the calibration is improved, some key metrics used in uncertainty estimation should be shown. For instance, the probability of the ground truth falls in the 1 - $\alpha$ predicted range.

- The entropy-based filter is not quite novel. Since the entropy here is in a monotonic relationship with $\hat{\sigma}_{calib}$. This finding just shows a common belief in the well-calibrated model that the more uncertain the model is, the greater error the model will make.

- The paper is quite mixed. It covers too many arguments which makes the outline of the paper unclear. Does the method focus on improving regression performance or uncertainty estimation? Some sections also include classification tasks, which makes the main idea of  the paper more vague.

---

> ### Author Response · Authors · 2023-06-11
> **Response to reviewer oWh4**
>
> We thank the reviewer for appreciating the problem tackled by our work and the proposed solution.
> Indeed, the KDE technique to reweigh the target for imbalanced regression is inspired by the analogous technique based on frequency of labels for imbalanced classification task.
>
> We revised our manuscript to reflect the suggestions pointed in the review (blue colour text in the revision). We address queries below:
>
> **Refining the flow and the main arguments**
>
> Our revisions better highlight our contributions and main arguments (text in blue) across various sections.
> In particular, abstract and introduction now clearly highlight that our method makes several contributions:
>
> * We introduce a KDE-based technique to tackle the imbalanced regression problem with medical time series, discussed in detail in Section-3.2.2 and 3.3.
> * Our method builds on probabilistic deep regression that allows uncertainty estimation for the predicted continuous values in the novel context of medical time series, presented in Section-3.2.3 and 3.3.
> * We propose a new approach for calibrating the predicted uncertainty in Section-3.3.2.
> * Finally, we demonstrate the efficacy of the calibrated uncertainty estimates to improve the predicted continuous value and also flag unreliable predictions, as discussed in Section-3.3.3.
>
> **Comparison with baselines trained with log/standardised targets**
> * We highlight that all the regression models in this study are trained with standardised targets, we have made this clearer in the manuscript in Section-4.2 (*revised in blue*).
> * We introduced *more baselines* for all the tasks (e.g., `Regres.-L1 (log)`, `Regres.-w.-U. (log)`, and `HypUC (log)` in Table-1) that are trained with standardised-log targets (Section-4.2 revised in blue). We observe that KDE-based reweighed standardised target trained models are superior to log-based targets (without KDE reweighing) in most cases, as shown in Table-1 (*rows in blue*). This indicates that to tackle imbalanced nature of the dataset, KDE based weighting scheme is more effective than transforming the targets alone.
>
> **More uncertainty calibration metrics**
> * In addition to the already presented regression calibration metric in Table-1 as discussed in **[a]** , thanks to this review we have now also included the suggested metric in the revised manuscript. As discussed in **[c,d]**, we quantify the calibration of the estimated uncertainty using the interval that covers the true value with a probability of $\alpha$, that is,
> $
> \mathcal{I}_{\alpha}(\hat{\mathbf{y}}, \hat{\mathbf{\sigma}}) = \left[ \hat{\mathbf{y}} - \frac{C^{-1}(\alpha)}{2} \hat{\mathbf{\sigma}}, \hat{\mathbf{y}} + \frac{C^{-1}(\alpha)}{2} \hat{\mathbf{\sigma}} \right]
> $.
> Where, $\hat{\mathbf{y}}, \hat{\mathbf{\sigma}}$ are predicted continuous values and uncertainty, and $C^{-1}(\cdot)$ refers to the quantile function, as discussed in Section-4.3 (*text in blue*).
> * We note that for good calibration $\alpha$ fraction of groundtruth samples should lie in the interval $\mathcal{I}_{\alpha}(\hat{\mathbf{y}}, \hat{\mathbf{\sigma}})$ and the length of the interval, $C^{-1}(\alpha) \hat{\mathbf{\sigma}}$, should be small. We report both metrics in our evaluation. For the interval length, we report the mean interval length of the test set, represented by $<len(\mathcal{I}_{\alpha}(\hat{\mathbf{y}}, \hat{\mathbf{\sigma}}))>$ (*blue column in Table-1*).
>
> **Entropy based filter for flagging unreliable prediction**
>
> * As pointed out in the review, indeed the entropy is a monotonic function of $\mathbf{\hat{\sigma}}_{calib}$. However, in the paper we wanted to point out that well-calibrated uncertainty estimates can potentially enable us to flag unreliable predictions. This may be particularly relevant since several existing AI/ML algorithms for clinical diagnosis with medical time series do not quantify the uncertainty in a prediction.
> * Our work is, to the best of our knowledge, the first one to demonstrate the potential applicability of the uncertainty estimates in a real-world clinical settings with time series data, as discussed in Section-4.3.3.
>
> **Comparing with classifiers**
>
> * Section-1,2 note that the current approach in literature is to perform classification even though diagnostic tests rely on continuous values. For instance, while serum potassium is a continuous number, current approach only classifies the potassium level as Hypo (low) / Normal / Hyper (high). In practice, Hyperkalemia has a spectrum and it is limiting to classify it as a single class.
> * To compare with such classification approaches, we note that the regression model also allows classification as described in this section-3.3.3 (*revised in blue*).
>
> [a] -  Well-calibrated regression uncertainty in medical imaging with deep learning, *MIDL 2020*
>
> [b] - Accurate Uncertainties for Deep Learning Using Calibrated Regression, *ICML 2018*
>
> [c] - A survey of uncertainty in deep neural networks - *by Jakob Gawlikowski et. al.*

---

> ### Author Response · Authors · 2023-06-14
> **Response to reviewer oWh4 - follow up**
>
> Respected reviewer,
>
> We are happy to address more of your queries, if there are any.
> We sincerely request you to let us know your thoughts.

---

### Review · Reviewer_hek2 · 2023-07-01

**Summary Of Contributions:**

This paper proposes HypUC, a framework for performing a regression task specifically on medical time-series data (e.g. ECG signal) while quantifying the prediction uncertainty. The paper proposes a modified objective function involving kernel density estimation to better cope with imbalanced regression, and introduces a post-hoc calibration method, namely Hyperfine Uncertainty Calibration. On a large-scale ECG dataset, the paper conducts four different regression tasks, showing that it outperforms standard regression methods. The paper also shows HypUC's efficacy regarding its calibrated-ness and effectively removing samples with uncertain predictions.

**Audience:**

Yes

**Claims And Evidence:**

No

**Requested Changes:**

After many revisions to reflect other reviewers' comments, I believe there is only one noticeable problem.
- The title says "medical time series" but the entire paper focuses on only one application, which is ECG-based prediction. Therefore, if the title is changed to "HypUC: Hyperfine Uncertainty Calibration with Gradient- boosted Corrections for Reliable Regression on Imbalanced Electrocardiogram", then there would be no problem. Or the paper could include more experiments with other medical time-series such as EEG, or vital signs, but I assume that would be too much to ask.

**Strengths And Weaknesses:**

Strengths:
- While targeting a very specific yet important task of regression with uncertainty, the proposed method is reasonably derived.
- The paper does not stop at simply proposing an objective function for regression, but addresses the practical aspect by proposing a calibration strategy as well as ensemble-based decision making
- The experiments are extensive in that HypUC was put to regression, calibration, and decision-making tasks, all showing its effectiveness.

Weaknesses:
- Although the title indicates HypUC is developed for general medical time-series, the entire experiment was conducted only on the ECG dataset.

---

> ### Author Response · Authors · 2023-07-01
> **Response to reviewer hek2**
>
> We sincerely thank the reviewer for acknowledging that our work targets a very important task with reasonably derived method, and that our experiments show the efficacy of derived calibrated uncertainty estimates in decision making.
>
> As suggested in the review we have now updated the manuscript to have the title:
>
> "HypUC: Hyperfine Uncertainty Calibration with Gradient-boosted Corrections for Reliable Regression on Imbalanced Electrocardiograms"

---

### Decision · Action_Editors · 2023-08-07

**Recommendation:** Accept as is

**Comment:**

Following the discussion, there was no controversy that this paper should be accepted. It is not on a topic that is popular at TMLR, however, it is well executed.

**Audience:**

This paper is a bit of an outlier for TMLR, however, it will be of interest to some.

**Claims And Evidence:**

The revisions answered all questions and concerns raised by the reviewers. This paper should be accepted.

---

> ### Author Response · Authors · 2023-08-27
> **Thank you all.**
>
> Respected Editors In Chief, Action Editors, and Reviewers,
>
> We thank you for your valuable comments that helped us improve the manuscript further.
> We have now uploaded the camera-ready version of the manuscript.
>
> Sincerely,
> Authors